# COVID-19 Detection on Chest X-ray and CT Scan: A Review of the Top-100 Most Cited Papers

**DOI:** 10.3390/s22197303

**Published:** 2022-09-26

**Authors:** Yandre M. G. Costa, Sergio A. Silva, Lucas O. Teixeira, Rodolfo M. Pereira, Diego Bertolini, Alceu S. Britto, Luiz S. Oliveira, George D. C. Cavalcanti

**Affiliations:** 1Departamento de Informática, Universidade Estadual de Maringá, Maringá 87020-900, Brazil; 2Instituto Federal do Paraná, Pinhais 83330-200, Brazil; 3Departamento Acadêmico de Ciência da Computação, Universidade Tecnológica Federal do Paraná, Campo Mourão 87301-899, Brazil; 4Departmento de Ciência da Computação, Pontifícia Universidade Católica do Paraná, Curitiba 80215-901, Brazil; 5Departamento de Informática, Universidade Federal do Paraná, Curitiba 81531-980, Brazil; 6Centro de Informática, Universidade Federal de Pernambuco, Recife 50740-560, Brazil

**Keywords:** COVID-19, pattern recognition, machine learning, chest X-ray, CT scan

## Abstract

Since the beginning of the COVID-19 pandemic, many works have been published proposing solutions to the problems that arose in this scenario. In this vein, one of the topics that attracted the most attention is the development of computer-based strategies to detect COVID-19 from thoracic medical imaging, such as chest X-ray (CXR) and computerized tomography scan (CT scan). By searching for works already published on this theme, we can easily find thousands of them. This is partly explained by the fact that the most severe worldwide pandemic emerged amid the technological advances recently achieved, and also considering the technical facilities to deal with the large amount of data produced in this context. Even though several of these works describe important advances, we cannot overlook the fact that others only use well-known methods and techniques without a more relevant and critical contribution. Hence, differentiating the works with the most relevant contributions is not a trivial task. The number of citations obtained by a paper is probably the most straightforward and intuitive way to verify its impact on the research community. Aiming to help researchers in this scenario, we present a review of the top-100 most cited papers in this field of investigation according to the Google Scholar search engine. We evaluate the distribution of the top-100 papers taking into account some important aspects, such as the type of medical imaging explored, learning settings, segmentation strategy, explainable artificial intelligence (XAI), and finally, the dataset and code availability.

## 1. Introduction

Since 2020, we have observed a significant amount of works published describing solutions for the most varied problems that arose due to the COVID-19 pandemic. As a consequence of technological development, many of these works present computer-based solutions to attack those problems.

Currently, a large number of medical imaging tests are performed every day because the digital image is quite suitable both for storage and also to support examination. At the same time, it is also widely known that digital imaging is the standard input for research developed by the pattern-recognition and machine-learning communities. Hence, we have faced a boom in the number of works published by these research communities devoted to supporting medical examination from medical imaging.

Since the beginning of the pandemic, pneumonia has been one of the most common consequences of COVID-19 due to the high level of exposure to the respiratory system. CXR and CT scans are the most commonly used imaging tests for diagnosing pneumonia, and CT scan is the gold-standard imaging test that best supports the analysis of the lungs. On the other hand, CXR is cheaper and more widespread around the world. Numerous studies have been developed by the pattern-recognition and machine-learning research communities specifically using these kinds of images. Figure 1 shows one example of each of these image types.

By searching for works already published in this context, we can easily find thousands of them addressing this topic from the most varied perspectives, such as pneumonia detection, pneumonia classification (in terms of the causative pathogen), lung region segmentation, infection region segmentation, and decision explanation. However, many of these works do not present a very impressive scientific contribution. In this way, here we describe a review of the top-100 most cited works published in the literature within this context according to the Google Scholar search engine (The search was carried out on 12 July 2022). The rationale behind this choice is that the number of citations obtained by a paper is probably the most straightforward and intuitive way to verify the impact of a given work on the research community.

In this review, we aim to address some important aspects related to the top-100 selected papers as the predominant computational methods used in this field of research. By analyzing the literature, we can find other reviews evaluating the top-cited COVID-19 papers. However, it is important to point out that, to the best of our knowledge, none of them explored thoracic medical imaging from the same perspective we did here, but from a more broadly oriented point of view [2,3,4].

This paper is organized as follows: Section 2 describes the study design and illustrates a taxonomy used to conduct the discussions along this work. Section 3 describes some details of the top-25 papers according to the number of citations. Section 4 is composed of specific subsections to discuss the top-100 papers taking into account aspects like type of medical imaging, type of learning, use of a strategy for segmentation, use of XAI, and dataset and code availability. In Section 5, concluding remarks are pointed out, and finally, Appendix A describes some information about the 75 papers not explored in Section 3.

## 2. Study Design and Taxonomy

This section describes how we organized the search for the works we discuss in this study. The search was performed by using the Google Scholar search engine on 12 July 2022. We decided to use this platform because it integrates works of all other scientific research portals (engines) and provides a reasonable estimate of the number of citations obtained by each work. The search was performed with the two following search queries: (i) (COVID AND (X-ray OR CT scan) AND (“image processing” OR “machine learning” OR “artificial intelligence” OR diagnosis OR detection)), and (ii) (COVID AND “deep learning”). In the former query, we have excluded works that do not present computer-based solutions, and we excluded results unrelated to CXR and CT scan solutions in the last query.

Following this, we performed the first filtering (F1), excluding works that correspond to reviews, surveys, or comparative studies, which do not correspond to our purposes. After this first filtering, we excluded a total of nine works that had been obtained in the first round. Thus, we took the subsequent nine most cited papers that do not belong to the category excluded in F1 to complete the top 100. Next, we performed a second filter (F2), excluding works that had not been peer-reviewed (preprints). After performing F2, we excluded a total of 18 works, and again, we took the subsequent 18 most cited studies that do not belong to the categories excluded both in F1 and F2 to complete the top-100.

Table 1 presents the top-100 most cited papers obtained after the first round and after each filtering. These details are (i) the average number of citations for the top-100; (ii) the h-index among the top-100 papers; (iii) the number of citations of the most cited paper; (iv) the number of citations of the least cited paper in the top 100.

Figure 2 shows a taxonomy containing the main aspects we considered for conducting the discussions in this study. We evaluated five aspects: (i) medical image, chest X-ray (CXR) or computed tomography (CT scan); (ii) learning approach, deep or shallow (we use the term shallow method to refer to any method other than deep learning); (iii) segmentation strategy, manual or automated using a deep network, common deep strategies includes U-Net [5], SegNet [6], and others; (iv) explainable artificial intelligence (XAI), common strategies includes class activation maps (CAM) [7], gradient-weighted CAM (Grad-CAM) [8], local agnostic linear model (LIME) [9], layer-wise relevance propagation (LRP) [10], and others; and, (v) dataset and code availability.

## 3. Overview of Top 25 Most Cited Papers

This section describes the main highlights of the top 25 most cited papers. We decided to restrict the number of works detailed, aiming to keep our list as short as possible while emphasizing its most important contributions. The selection of the top 25 most cited papers is purely quantitative and does not consider any particular characteristic. Assuming that the number of citations is a metric for scientific quality and importance, it is interesting to describe the most cited papers to find out exactly what they proposed and evaluated to achieve popularity in such a short term.

Wang et al. [11] presented the most cited paper found in our search protocol, described in Section 2, with a total of 1848 citations. In that work, the authors performed COVID-19 detection by using the COVID-Net, a deep convolutional network specially tailored to detect COVID-19 from CXR images. The developed network is open source and was made available to the general public. The authors also introduced COVIDx, an open access dataset composed of 13,975 images obtained from 13,870 patients, probably with the largest number of positive cases available at that moment. The dataset was created by taking images from other sources of CXR images. In addition, the authors used an explainability method to aid clinicians in improving the screening process, adding transparency and reliability to the provided results. The work attracted much attention, probably for the following reasons: it was one of the first open-source networks designed for COVID-19, it made available a quite useful dataset with a significant number of positive exams, and finally, it was published at a very opportune time, in 2020.

Ozturk et al. [12] addressed COVID-19 detection from CXR images by using the DarkNet model as a classifier for the you only look once (YOLO) real-time object-detection system. The work is the second-most cited paper in the list obtained in our review, with a total of 1523 citations. The problem was addressed both as binary (COVID-19 vs. no findings) and multi-class classification (COVID-19 vs. no findings vs. pneumonia). The classification accuracy obtained was 98.08% for binary classification and 87.02% for multi-class. The authors implemented 17 convolutional layers in the model, including different filtering on each layer. The model was made available on GitHub. The main positive remarks of this work were the impressive moment when it was published, in April 2020, the evaluation of the results by radiologists, and also the availability of the model to the public. However, the authors admit that the work was done with a limited dataset, and future improvements on a more robust dataset should be pursued.

Apostolopoulos et al. [13] experimented with automatic COVID-19 detection from X-ray images by using convolution neural networks with transfer learning. For this, the authors composed two datasets (i.e., Dataset_1 and Dataset_2) by using images taken from three different sources: (i) the collection of X-ray images of Professor Joseph Cohen from the University of Montreal; (ii) a set of X-ray images obtained from websites such as the Radiological Society of North America, Radiopaedia, and the Italian Society of Medical and Interventional Radiology; (iii) and finally, a collection of common bacterial–pneumonia X-ray scans was included, to train the model to distinguish COVID-19 from other types of pneumonia. Dataset_1 was composed of 224 positive COVID-19 images, 700 bacterial pneumonia images, and 504 healthy lungs images. Dataset_2 was composed of 224 positive COVID-19 images, 504 healthy images, and 714 images of both bacterial and viral pneumonia (400 bacterial and 314 viral). The images were all resized to 200 × 266, and they were evaluated by using the following models: VGG19, MobileNetV2, Inception, Xception, and Inception ResNet v2. The fine-tuning was performed separately for each model evaluated, so each one had its own parameters defined. The training and evaluation were done by using 10-fold cross-validation, and the best results were obtained with the MobileNet v2 model, which achieved an accuracy of 96.78% and 94.72 for binary and three classes classification, respectively.

Narin et al. [14] performed COVID-19 detection from X-ray images by using five convolutional neural network models and three different public datasets (i.e., Dataset_1, Dataset_2, and Dataset_3). Dataset_1 is composed of 341 X-ray images obtained from Dr. Joseph Cohen’s open source GitHub repository, Dataset_2 has 2800 healthy chest X-ray images from the ChestX-ray8 database, and Dataset_3 is made of 2772 bacterial and 1493 viral pneumonia chest X-ray images from the Kaggle Chest X-Ray Images (Pneumonia) repository. Five pre-trained models were used in this work: ResNet50, InceptionV3, ResNet101, Inception-ResNetV2, and ResNet152. The authors performed their experiment by using three different binary classes: Binary Class-1 (COVID-19 vs. healthy), Binary Class-2 (COVID-19 vs. viral pneumonia), and Binary Class-3 (COVID-19 vs. bacterial pneumonia). The evaluation used a five-fold cross-validation and the ResNet50 pre-trained model obtained the bests results, with an accuracy of 96.1% in Binary Class-1, 99.5% in Binary Class-2, and 99.7% in Binary Class-3. The highlights of this paper include the fact that it used more data than many other articles at the time it was published, as well as its significantly high performance.

Wang et al. [15] hypothesized that by analyzing CT scan images taken from the lungs, it is possible to extract graphical features of COVID-19 providing a clinical diagnosis ahead of the pathogenic test typically made by laboratories. Thus, the authors performed experiments on a dataset composed of 1065 CT scan images of COVID-19 confirmed patients and others taken from typical viral pneumonia. Three Chinese hospitals provided the images. In the proposed pipeline, the authors first performed some preprocessing of the images and manually delineated the regions of interest (RoIs) on the images. Transfer learning was done by using a predefined model (i.e., GoogleNet Inception V3) already trained on 1.2 million images from ImageNet labeled into 1000 categories. The authors proposed a modified inception (M-inception) for classification by changing the last of the fully connected layers. The feature’s dimensionality was reduced before it was sent to the final classification.

Finally, the authors performed a robust evaluation of the method, addressing some critical points closely related to the practical feasibility of the employment of the proposal. For the performance evaluation, the authors first trained and tested the system, exclusively using images from the same hospital. In this scenario, the accuracy rate was 89.5%. Next, another round of experiments using images from the three hospitals was performed, and the obtained accuracy was 82.5%. Another important comparison was between the results obtained by the system and radiologist prediction. Two radiologists assessed the images and achieved an accuracy of approximately 55%. This result demonstrates the advantage of the use of the proposed method. Lastly, the authors experimented on 54 images incorrectly predicted (false negatives) by using nucleic acid testing, the gold standard for COVID-19 diagnosis. The system was able to predict 46 out of the 54 images correctly.

Xu et al. [16] sought to develop an early screening model to detect COVID-19 from pulmonary CT images by using deep learning techniques. The dataset used contained 618 transverse-section CT samples (219 COVID-19, 224 Influenza-A viral pneumonia, and 110 healthy) provided by three Chinese hospitals. In the first step of their approach, the authors preprocessed the CT images to select the most effective pulmonary regions. Afterward, a total of 3957 candidate image cubes were segmented by a 3D CNN segmentation model; because the cube’s middle region contained the most amount of information about the infection, the cube center image and its two neighbors were selected, totaling 11,871 image patches used for training and classification. The authors evaluated two models: a traditional ResNet-18 based model and a ResNet-18 model concatenated with a location–attention mechanism. In the first step of the evaluation, the authors tested the classification for a single image patch; ResNet-18 achieved an accuracy rate of 78.5% whereas ResNet-18 plus location attention mechanism achieved 79.4%. Because of the lower performance, the ResNet-18 model was not used in further experiments, The authors then analyzed the classification of CT samples as a whole, and an overall accuracy rate of 86.7% was achieved by the ResNet-18 plus location attention mechanism.

Khan et al. [17] introduced the CoroNet, a deep convolutional network specially designed for COVID-19 detection from CXR images. The model was based on Xception architecture pre-trained on the ImageNet dataset and end-to-end trained on an image collection curated for the development of the study. The model was evaluated on two different scenarios, the first considering four classes (COVID-19 vs. pneumonia bacterial vs. pneumonia viral vs. normal), obtaining an accuracy of 89.6%, and the second with three classes (COVID-19 vs. pneumonia vs. normal), achieving an accuracy of 95%. The work was presented in May 2020, at the pandemic’s beginning. One of the main contributions of the work was to point some directions and to indicate that deep models could adequately be used to address COVID-19 detection from CXR images with minimum pre-processing of data. In addition, the authors also claim that further improvements could be achieved with more extensive sets of data.

To overcome the limited availability of annotated medical images in the context of COVID-19 diagnosis, Abbas et al. [18] experimented with a deep CNN called decompose, transfer, and compose (DeTraC) for COVID-19 classification from CXR images. DeTraC is intended to properly deal with irregularities present in the dataset by using a class decomposition mechanism to investigate its class boundaries. For this, a class composition layer is introduced to clarify the final classification. A class decomposition component is included before the knowledge transformation from an ImageNet pre-trained CNN model, and a class composition component is included after that. The model was evaluated on a comprehensive dataset composed of images taken from several hospitals worldwide. An accuracy of 93.1% was obtained in detecting COVID-19 from normal and severe acute respiratory syndrome cases.

Song et al. [19] developed a deep learning-based CT diagnosis system evaluated on a dataset composed of CT scan images obtained from 88 patients diagnosed with COVID-19, 100 patients infected with bacterial pneumonia, and 86 healthy persons for comparison and modeling. The proposed system was very successful in detecting the primary lesions present in CT images. Moreover, we can highlight that the work was developed at a very early stage of the COVID-19 pandemic. Although the work was published in March 2021, the first version of the manuscript was submitted in April 2020, when very few positive COVID-19 images were available. The deep learning solution proposed was based on three main steps: first, the central region of the lung is extracted. Next, a details relation extraction neural network (DRENet) was used to obtain image-level predictions. Finally, the image-level predictions were aggregated to obtain the person’s diagnosis. The model could discriminate COVID-19 from bacterial pneumonia with a recall of 0.96. For discriminating COVID-19 from bacterial pneumonia and healthy persons, the system obtained a recall of 0.95. The authors made the system available for COVID-19 diagnosis at an online server, and the source codes and the datasets were also made available. However, one drawback of the proposal is that it could not keep good prediction rates when evaluated on external data.

Oh et al. [20] also investigated COVID-19 features on CXR images at an early stage of the pandemic. At that moment, there was a huge scarcity of data. Thus, the authors proposed a patch-based CNN approach with a relatively small number of trainable parameters. The method was based on the use of statistical analysis of the potential biomarkers of CXR images. The first step in the proposed general framework is the data normalization, as a pre-processing stage. In addition, a segmentation network is used to isolate the lung areas as regions of interest. Then, patches are obtained from the lung area and used for training a classification network. For testing, the decision for each image is based on the majority voting involving the decisions taken for the patches created from the image. The experimental results demonstrated that the method was able to get state-of-the-art performance.

Ardakani et al. [21] experimented with 10 convolutional neural networks to evaluate the application of deep learning techniques in routine clinical practice. The authors used CT images from 194 patients (108 COVID-19 and 86 non-COVID-19) in their study. The region of interest of the images—that is, the region of infection—were segmented manually by an expert. After being segmented, they were cropped and resized to 60 × 60 pixels. The authors used various performance metrics to evaluate their and a diagnosis from a radiologist expert. The best results in accuracy were found for the ResNet-101 (99.63%) and Xception (99.38%) networks; however, ResNet-101 was able to diagnose COVID-19 with higher sensitivity when compared to Xception, which is highly desirable when diagnosing diseases. When analyzing a single image patch, the radiologist was no match for the CNNs; he achieved better performance when analyzing the whole CT slice, but his accuracy was still lower than most CNNs. Overall, the authors successfully created a computer-aided diagnosis with promising results.

Chen et al. [22] developed a study proposing a deep model for COVID-19 detection from high-resolution CT scan images. For this purpose, the authors curated a collection composed of 46,096 images obtained from 106 patients at the Renmin Hospital of Wuhan University (China). Among these patients, 51 were affected by COVID-19 pneumonia, confirmed by laboratory test. The other 55 were control patients of other diseases. The model was built on top of U-Net++ architecture, having all the parameters loaded from ResNet-50 pre-trained on the ImageNet dataset. An external evaluation was conducted at another hospital to verify the system’s robustness, and it obtained an accuracy of 98.85% per image and 95.24% per patient. The system performance was also compared to expert radiologists on data from 27 prospective patients of Renmin Hospital. The system performance was considered comparable to the human expert’s performance, and, in addition, the time consumed by humans to perform the evaluation assisted by the system decreased by 65%. Lastly, it is worth mentioning that this study was published in November 2020.

Ucar et al. [23] proposed a new model for diagnosing COVID-19 based on deep Bayes–SqueezeNet called COVIDiagnosis-Net. The authors used chest X-ray images from the available CovidX dataset to train their model. This dataset is made of three classes: normal, pneumonia, and COVID-19. Compared to the other two classes, there are few images of COVID-19 in the dataset. Therefore, the authors performed a detailed offline augmentation over the COVID-19 class to overcome the imbalance ratio. The authors proposed model reached an accuracy rate of 98.3%, better performance compared to the state-of-the-art methods at the time (early 2020). In addition, the COVIDiagnosis-Net is significantly smaller than other models, such as AlexNet, hence, being ideal for implementation in embedded and mobile systems.

Afshar et al. [24] proposed a capsule network, called COVID-Caps, aiming to circumvent the difficulty of CNNs in dealing with spatial information between different instances of the image. The proposed capsule network presents four convolutional layers and three capsule layers. The network is fed with 3-D X-ray images. The loss function was also modified to deal with the class imbalance issue. At that moment of the pandemic, obtaining enough data for experimentation was not easy. The model was capable of obtaining an accuracy of 95.7%. In addition, aiming to get better results, the authors experimented with training and transfer learning based on an external dataset. Differently from other works, the authors conducted the pre-training by using X-ray images. Following this protocol, the authors obtained an accuracy of 98.3%. Lastly, it is essential to mention that the model was made available publicly for open access.

Panwar et al. [25] proposed a deep learning neural network method called nCOVnet to create an alternative fast screening method for detecting COVID-19. The authors used Dr. Joseph Cohen’s open source GitHub repository and Kaggle’s Chest X-Ray Images (pneumonia) as the dataset. Data augmentation techniques were applied to overcome the dataset limitations. It is worth noting that the authors took extra precautions to prevent data leakage. The nCOVnet uses the VGG16 model as the base layer of the architecture and adds to it five custom layers as a head model. In conclusion, nCOVnet could predict COVID-19 from CXR images with 97.97% confidence.

Huang et al. [26] presented a quantitative evaluation of burden changes in COVID-19 patients by using a deep learning method on serial CT scan images. The method was based on the evaluation of a quantitative image parameter (called QCT-PLO), automatically generated by a deep learning software tool from chest CT scans. The authors concluded that the quantification of lung opacification was significantly different among COVID-19 patient groups with different levels of severity. In conclusion, they claim that this method could eliminate the subjectivity in the initial assessment and follow-up of pulmonary findings for COVID-19.

Togaçar et al. [27] performed COVID-19 detection from CXR with a dataset containing three classes: COVID-19, pneumonia, and healthy. The data classes were restructured by using Fuzzy Color, an image-stacking technique. Image stacking combines multiple images, aiming to improve the quality of the images in the dataset, eliminating noises from them. The deep learning models MobileNetV2 and SqueezeNet were used to train the stacked dataset and the feature sets obtained were processed by using the Social Mimic Optimization (SMO) method. Lastly, efficient features were combined, and the classification was performed by using support vector machine (SVM). The overall classification rate obtained was 99.27%. The authors claim that the proposed preprocessing enhanced the feature extraction efficiency by using the SMO algorithm. In addition, they also demonstrate the usability of the proposed approach in mobile devices.

Pereira et al. [28] investigated COVID-19 identification from CXR images considering different perspectives. In the first scenario, the authors performed multiclass classification by using CXR images containing pneumonia caused by different pathogens (COVID-19, SARS, MERS, streptococcus, and pneumocystis). Then, the authors identified a hierarchy between the different pathogens and investigated the classification considering a hierarchical scenario. The authors also experimented with resampling algorithms to deal with the natural imbalance between the different types of pneumonia. In addition, a dataset (named RYDLS-20) was composed of publicly available datasets. Lastly, it is important to mention that they also experimented with the use of handcrafted features by evaluating a comprehensive set of texture descriptors and non-handcrafted features, automatically obtained by using deep learning models. The best result obtained for COVID-19 was found in the multiclass scenario, with an F-Score of 0.89. The paper was published at the beginning of May 2020.

Wang et al. [29] presented a fully automated deep learning system to diagnose COVID-19 and stratify patients into high- and low-risk groups. The authors used a large dataset with 5372 computed tomography exams. The dataset was collected from various cities or provinces of China. To acquire the lung mask of the CT images, the authors performed lung segmentation by using the DenseNet121-FPN deep learning method. After that, non-lung tissues and organs that may still exist in the region of interest were suppressed. For the diagnosis and prognosis, the researchers used their proposed model, COVID19Net, which uses a DenseNet-like structure. The training of COVID19Net was performed in two steps: (i) train the model with a large dataset (4106 patients) of lung cancer; (ii) transfer the pre-trained model to the COVID-19 dataset. For prognostics, the authors combined the 64-dimensional feature from the COVID-19Net and combined it with clinical features (age, sex, and comorbidity). This new feature vector was used to build a multivariate Cox proportional hazard model. COVID-19Net reached an AUC of 0.90 in the training set and obtained similar results in two other validation sets, 0.87 and 0.88, respectively. Regarding the prognostic, Kaplan–Meier analysis showed that patients classified in the high-risk group had a higher hospital stay time when compared to the low-risk group.

The lack of a publicly available dataset with CXR and CT scan images is one of the biggest obstacles that obstruct the research of COVID-19 artificial intelligence-based solutions. Aiming to circumvent this problem, Maghdid et al. [30] presented a comprehensive dataset of both these types of images, obtained from multiple sources. The dataset comprised 170 CXR images and 361 CT scan images in its first version. In addition, they also presented a simple CNN and modified pre-trained AlexNet model and experimented on CXR and CT scan images. The experimental results achieved an accuracy of up to 94.1% by using the first model and up to 98% by using the latter.

Brunese et al. [31] presented a two-step approach to detect COVID-19. The authors created two deep learning models. The first model can distinguish between healthy X-ray chest images and those showing individuals with pulmonary disease. If the X-ray image is labeled as pulmonary disease, a second model detects whether the pulmonary disease is pneumonia or COVID-19. In addition, the researchers used the GRAD-CAM activation map to highlight the most significant areas in the COVID-19 detection. Brunese et al. models were based on the VGG-16 model and used transfer learning methods. The dataset used in this work combines three others, two of which are freely available datasets. It contains 6523 X-Ray images in total. Regarding the evaluation of the models, the first one (healthy vs. pulmonary disease) obtained an accuracy and sensitivity of 0.96, and the second one (pneumonia vs. COVID-19) reached an accuracy of 0.98 and a sensitivity of 0.87.

Loey et al. [32] evaluated COVID-19 detection by using deep transfer learning and generative adversarial networks (GAN) to apply data augmentation. The experimental dataset was composed of images taken from other publicly available datasets. The authors experimented with three different scenarios: four classes (COVID-19, normal, viral pneumonia, and bacterial pneumonia), three classes (COVID-19, normal and bacterial pneumonia), and finally, binary classification (COVID-19 vs. normal). Reasonable performance rates were obtained in the aforementioned scenarios: for four classes, the best rate obtained was 80.6% of accuracy, for three classes, 85.2% of accuracy, and on the binary classification, 100% accuracy was obtained. Despite the impressive results, the code and dataset used in this work were not made publicly available.

Islam et al. [33] performed COVID-19 classification from CXR images by using CNN to make feature extraction and long short-term memory (LSTM) for classification. The experiments were carried out on a dataset created by using images from publicly available collections containing positive COVID-19 CXR images. The images were divided into three classes: COVID-19, normal, and pneumonia (other than COVID-19). As a result, the authors obtained an accuracy of 99.4%, and they claim that the proposed CNN-LSTM architecture overcame the results obtained by using a competitive CNN architecture.

Ismael et al. [34] experimented with shallow and deep learning approaches to detect COVID-19 in chest X-ray images. The authors used a dataset with 180 COVID-19 and 200 normal CXR images in their experiments. Regarding the deep learning approaches, fine-tuning procedures were done for the ResNet18, ResNet50, ResNet101, VGG16, and VGG19 models. Furthermore, an end-to-end CNN model was trained. As for the shallow approach, Ismael et al. evaluated the SVM classifier trained with deep learning features and various texture extractors such as LBP, LPQ, BSIF, and others. Overall, the deep learning approaches outperformed the local descriptors. The best result was achieved by combining ResNet50 features with the SVM classifiers; this combination achieved an accuracy of 95.79%. Other approaches are also worth mentioning: fine-tuning of ResNet50 achieved an accuracy of 92.6%, end-to-end training of CNN achieved an accuracy of 91.6%, and BSIF achieved an accuracy of 90.5%.

To aid in the screening of COVID-19, Amyar et al. [35] proposed a multi-task deep learning (MTL) approach. The proposed MTL architecture was based on three tasks: COVID-19 vs. normal vs. other infections classification, COVID-19 lesion segmentation, and image reconstruction. The authors collected CT images from three different datasets, totaling 1369 CT scans for their study. The performance of the MTL was compared with various state-of-the-art models, including U-NET for segmentation and Alexnet, VGG-16, VGG-19, ResNet50, and others for classification. The MTL performed significantly better than the state-of-the-art approaches in both segmentation and classification. In the COVID-19 lesion segmentation task, the MTL achieved an accuracy of 95.23%, whereas U-NET achieved 83.40%. As for classification, the proposed method had an accuracy of 94.67%, whereas the best among the state-of-the-art tested models had an accuracy of 90.67%.

Table 2 presents some of the most remarkable details about the 25 papers described in this section. The remaining 100 papers evaluated in this study are listed in Table A1, presented in the Appendix A.

## 4. General Statistics

This section describes some statistics that can help to identify how the papers investigated in this review are distributed, considering some important aspects in which they can be categorized.

### 4.1. Citations

The number of citations that early COVID-19 papers received is significant. Nowadays, after approximately two years after the pandemic started, the top paper in this review was cited by 1848 subsequent works. Usually, such a number is obtained after many years of publication. All of that makes the COVID-19 pandemic a significant event worth analyzing.

The number of citations ranged from 1848 to 65, with a mean of 251.5 citations, a standard deviation of 323.7, and an interquartile range of 182.7.

Since the time of the publication is critical, to normalize the number of citations, we calculated the average number of citations per day (CPD) for each of the 100 papers and sorted them in decreasing order. Only two papers originally out of the top 25 made their way into this list. In general, only a few slight changes affected the original ranking. The work carried out by Kassania et al. [36] was originally in the 35th position, and after reordering, it was placed in the 16th position; the work presented by Rahman [37] was originally placed in the 32nd position, and after reordering, it was placed in the 20th position.

### 4.2. Publication Dates

As discussed, COVID-19 attracted researchers from many different areas, resulting in many published works quickly. A simple search in any engine can easily retrieve hundreds of published papers in different fields.

Given the amount of popularity around the topic, the dates of submission and publications and their difference are fascinating detail to analyze. At first, it is possible to notice a constant flow of publications from the beginning of the pandemic until the second quarter of July 2021. Figure 3 presents an exploratory overview of the top-100 papers selected and submission and publication dates. For 14 papers, the submission date is unavailable; hence, only the publication date is displayed. Among the selected papers, only one was submitted and published after June 2021. The reason for that might be simply that we selected papers based on the number of citations; there was probably not enough time for the newer papers to obtain the proper amount of citations.

Table 3 displays the exact number of papers submitted and published per quarter from 2020 Q1 to 2022 Q1. Among the selected papers, most were submitted during the first half of 2020, and almost all within the first and third quarters of 2020. Considering 86 papers with submission dates available, 60 (≈70%) were submitted in the first half of 2020, and 78 (≈91%) were submitted in the first three quarters of 2020.

Such a skewed distribution of early submissions is somewhat expected for two main reasons: (i) there was a huge commotion at the start of the pandemic to find solutions that could be applied in practice, and (ii) early papers laid the foundations by proposing novel datasets and methods, and hence obtained many citations by subsequent works.

In this vein, it is also possible to notice when analyzing the submission and publication time difference in Figure 3 that many papers had a minimal time difference, meaning that editors and publishers were very fast and efficient in publishing early COVID-19 related papers.

### 4.3. Countries

The top-100 papers published originated from 34 countries in total. We considered each paper’s author’s institution country for the analysis. Figure 4 presents an exploratory visual representation of their global distribution. China dominated the publication share with a total of 207 authors, 32.6% of the total. India followed them with a total of 65 authors, 10.2% of the total. The United States is in the third position with 42 authors, 6.6%. Appendix B presents the exact distribution for all 34 countries.

### 4.4. CXR vs. CT Scan

Two prominent medical image tests are used to investigate the lungs and, consequently, to support pneumonia diagnosis: CXR and CT scan. Even though CT scan is considered the gold standard for pneumonia analysis, we cannot ignore that CXR has many advantages as well, as it is more widespread, cheaper, and faster to obtain. There are several health centers worldwide where a CXR machine is available and a CT scan machine is not.

CXR images were the most frequently used in the top-100 papers reviewed here, exclusively assessed in 61 of them. Furthermore, 28 papers used only CT images, and 11 used both these types of images, as shown in Table 4. As discussed in Section 4.9, the likely reason for such distribution is twofold: (i) there were many COVID-19 CXR datasets available early, and (ii) CXR images are much more manageable and lighter to process than a CT scan volume. The average number of citations per paper did not vary much between CT-scan and CXR images.

### 4.5. Datasets

The limited number of publicly available CXR and CT scan images was a shortcoming of almost every paper reviewed here. We have to remember that most top-cited papers were published in 2020 at a time when any information around COVID-19 was still being published in the early days of the virus.

Table 5 presents the most frequent datasets used. Many papers composed novel datasets by combining multiple images from different sources. Such a trend is clear when we analyze the usage frequency of each dataset. For instance, Kaggle (pneumonia) and ChestX-ray8/ChestX-ray14 are publicly available datasets that precede the pandemic. Researchers are using them to extract images from other pathogens or healthy patients.

The Dr. Joseph Cohen initiative was the most used dataset throughout with 55 papers. It was followed by two non-COVID datasets, Kaggle (pneumonia) and ChestX-ray8/ChestX-ray14 with 29 and 22 papers using them, respectively.

The availability of public datasets is a game changer. Table 6 presents the distribution of public and private datasets. Most of the selected papers used public datasets for evaluation and received substantially more citations.

### 4.6. Learning Setup

The learning setup is the set of decisions, details, and parameters that control the classification process, comprised mainly of algorithms and data transformations.

In this review, we are separating the classifier type into two categories: deep and shallow methods. We use the term shallow method to refer to any method other than deep learning. Table 7 presents the classifier type distribution in this review. The use of deep learning methods, especially convolutional neural networks (CNNs), has been steadily increasing and dominating pattern-recognition tasks based on images over the last few years. The trend is very prominent in the selected papers: 81 exclusively applied deep learning, whereas only 12 applied shallow methods, and 7 used both machine learning methods. The average number of citations per paper is also substantially more significant in deep learning proposals.

One of the main advantages of CNNs, compared to shallow methods, is their ability to automatically learn helpful features from images, reducing the burden of applying and evaluating multiple handcrafted feature extractors. However, deep learning requires a large amount of training data to converge due to many trainable parameters.

Usually, it is unfeasible to use shallow methods with images directly, as it is performed with CNNs. First, one must extract features from the images by using handcrafted methods. Handcrafted methods can summarize image characteristics, such as texture, shape, color, and others. The weights from a layer of a pre-trained CNN can also be used as features, and they are referred to as deep features. The deep features extraction is an automated process, i.e., it does not focus on a specific characteristic. The usage of deep features aims to take advantage of CNNs ability to learn valuable features automatically while reducing the need for a large dataset, which is precisely the case of COVID-19 data scarcity, especially during the pandemic’s early days.

As shown in Table 8, among the 19 papers using shallow methods, eight (≈42%) used deep features, five (≈26%) used handcrafted features, and six (≈32%) leveraged both kinds of features.

Nevertheless, several techniques have been proposed to overcome the deep learning hunger for data. Transfer learning and data augmentation are probably the most popular among them.

Transfer learning is a method that uses the knowledge obtained from solving one problem to another different problem as a starting point. Then, the model could be fine-tuned for the specific task. As displayed in Table 9, in the papers reviewed, 65 used transfer learning while 32 did not, and two evaluated models with and without it. ImageNet was the most frequent problem used as a starting point, given its ability to generalize well in many subsequent tasks, even on medical tasks [38]. X-ray and CT scan images are visually and perceptually different from the images available in ImageNet, which could ultimately render the transfer learning useless. However, there are reports in the literature showing that even in this setting, the transfer learning from ImageNet can boost the performance across various deep models [39].

Data augmentation is a technique used to increase the training data available by slightly changing the already existing data. The transformations include rotations, translations, crops, random changes in color, brightness, contrast, and other factors. The creation of synthetic data is also considered a type of data augmentation. A generative adversarial network (GAN) has been applied to generate synthetic images. Transfer learning also helps to reduce overfitting, acting as a model regularizer. Amid the papers reviewed, Table 10, 48 used data augmentation when training, whereas 51 did not, and one evaluated both scenarios. Again, given the data scarcity of COVID-19 images, data augmentation could be a powerful ally when training deep models.

### 4.7. Segmentation

In digital image processing, image segmentation separates the input into multiple segments to ease subsequent analysis or further processing. In classification tasks, segmentation might reduce the unnecessary image background information that can interfere with the recognition process. In medical image analysis, segmentation could be considered an even more essential task because a misdiagnosis can have severe consequences for a patient following unfair treatment.

Considering a scenario of COVID-19 identification by using medical images, one would ideally first segment the lung area to remove the unnecessary information and then perform the detection or classification. The rationale is straightforward, the inflammation caused by COVID-19 is located in the lung area, so isolating it would only improve the classification.

As displayed in Table 11, among the top-100 papers analyzed in this review, only 25 considered lung segmentation as a part of the classification pipeline. The 75 remaining skipped it entirely and did not discuss its reason or justification. Out of the 25, two papers manually segmented the RoI before proceeding to the detection [15,21]; the 23 remaining articles applied automated strategies, usually based on deep networks, to segment the lung region.

Many people reason that by using deep strategies, we can overlook some pre-processing steps, such as segmentation, due to the amount of data available. However, thinking critically about segmentation has significant benefits that cannot be ignored. Considering COVID-19, there are reports in the literature showing that without segmentation, the model might be focusing outside the lung region, resulting in a biased performance [40,41,42]. Hence, the classification performance obtained in works that did not apply lung segmentation could also be biased.

### 4.8. Explainable Artificial Intelligence (XAI)

Explainable artificial intelligence (XAI) is a field that focuses on methods and approaches that can be utilized to explain model predictions. The primary objective is to determine which features the model actively employs when making predictions. When training deep models, there is no assurance as to which feature the model will prioritize, which is why such models are frequently referred to as black-box classification models.

Often, XAI can be used to verify which portions of the input image are being decisively used to reach a particular prediction. In medical images, it is possible to take advantage of such behavior to ensure the model focuses on the right things.

Following almost the same trend as segmentation, of the 100 papers considered, only 25 applied XAI to evaluate the black-box model. Table 12 presents the distribution of each XAI method used in featured papers. As some applied more than one XAI method, the total number is above 100. Methods based on class activation mapping (CAM) and its variations are the most popular, most likely due to their simplicity, ease of use, and overall accuracy [7,8,43]. Other interesting visualization methods are also used less frequently, including saliency maps [44,45], LIME [9], LRP [10], and GSInquire [46]. One of the papers applied a proprietary software called the uAI Intelligent Assistant Analysis System to analyze CT scans [47].

### 4.9. Reproducibility

For obvious reasons, the concern with the quality and reproducibility of the works addressed here cannot be left out. This section discusses the availability of datasets and codes among the selected papers.

We cannot neglect that the pandemic brought several critical factors that made the research development much more difficult. At the first moment, the scarcity of data was one of these factors. When the pandemic arose, naturally, there was not enough available and labeled data to support the COVID-19-related research development.

In this context, some researchers put much effort into providing, as fast as possible, datasets with labeled and organized data that could be made available to the research community.

Here, we list some of the most remarkable pioneer initiatives for COVID-19 dataset creation, both for CXR- and CT-scans:COVID-19 image data collection (https://github.com/ieee8023/covid-chestxray-dataset (accessed on 12 July 2022)), created by Cohen et al. [1].COVID-19 DATABASE (https://sirm.org/category/senza-categoria/covid-19/ (accessed on 12 July 2022)), made available by the Italian society of medical and interventional radiology.COVID-19 Dataset (https://www.kaggle.com/datasets/tawsifurrahman/covid19-radiography-database, https://www.kaggle.com/datasets/prashant268/chest-xray-covid19-pneumonia (accessed on 12 July 2022)), available in Kaggle.COVID-CT-Dataset: A CT Scan Dataset about COVID-19 (https://arxiv.org/abs/2003.13865 (accessed on 12 July 2022)).SARS-CoV-2 CT scan dataset: A large dataset of real patients CT scans for SARS-CoV-2 identification (https://www.medrxiv.org/content/10.1101/2020.04.24.20078584v3 (accessed on 12 July 2022)).

The datasets mentioned above were of great importance to the scientific developments obtained in this field of research during the pandemic. However, making some remarks regarding use of the datasets in most investigations is crucial.

The vast majority of works argue that the scarcity of data was an obstacle to experimental development. In this sense, many works performed experiments on particular image collections, sampling the datasets mentioned earlier and other non-COVID image datasets created before the pandemic to investigate the occurrence of other lung diseases, such as cancer. Among these non-COVID datasets, we can remark the chest X-ray dataset (https://www.kaggle.com/datasets/nih-chest-xrays/sample (accessed on 12 July 2022)), was composed of images provided by the National Institutes of Health (NIH), an American medical agency. This dataset was first introduced to community research by Wang et al. [48]. Other important sources used in many works are the Radiological Society of North America (RSNA) data collection (https://www.kaggle.com/c/rsna-pneumonia-detection-challenge (accessed on 12 July 2022)), and the Radiopaedia imaging datasets (https://radiopaedia.org/articles/imaging-data-sets-artificial-intelligence (accessed on 12 July 2022)).

On the one hand, creating ad hoc datasets was a good strategy to overcome the limitations imposed by the lack of data, favoring the creation of more robust models. On the other hand, it makes it very difficult to compare the results obtained by different works directly. Hence, we do not focus on the classification rates obtained by the works reviewed here; furthermore, different works do not necessarily use the same metric for performance evaluation. It is also essential to observe that in some works, the authors organized the data to perform binary classifications (e.g., COVID-19 vs. non-COVID-19), whereas in others, a multi-class scenario was proposed (e.g., COVID-19 vs. bacterial pneumonia vs. viral pneumonia vs. normal).

Code availability is another important aspect related to reproducibility. Code availability is of great importance for continuous research development, as it can allow other researchers to search for progress starting from other works previously developed. Approximately one in three works contribute in this sense. Thirty-three papers among the top 100 made the codes available. This rate is slightly better among the top-25 papers; there are 11 papers with codes made available among them (44%).

### 4.10. Non-Peer-Reviewed Excluded Papers

As already mentioned, we have excluded papers that were not peer reviewed in the second filter round (F2). We decided to exclude these papers, aiming to ensure some reasonable level of quality, validity, and originality. However, the fact that those papers were not peer-reviewed does not necessarily imply low quality, as suggested by their impressive number of citations. Table 13 summarizes the 18 works excluded in F2. The best-ranked paper in this list (Hedman et al. [49]) was placed in the eighth position before filtering.

## 5. Concluding Remarks

First of all, it is important to point out that all the efforts done looking for COVID-19 solutions are worth noting, and many vital achievements were obtained thanks to the commitment of the research community from different fields of study. However, it is also reasonable to look back and evaluate the significant impacts and contributions in the context investigated here and some limitations that may have obstructed achieving even better results.

Based on the rationale that the number of citations obtained by a paper is probably the most straightforward and intuitive way to verify its impact on the research community, we described here a review on the top-100 most cited papers considering the development of computer-based strategies for COVID-19 detection from thoracic medical imaging. Following, we highlight some remarkable findings, and we analyze them from the different perspectives addressed in this review.

One of the first aspects that attract attention is the vast majority of deep learning methods compared to shallow methods. On the one hand, this makes sense because deep models have been getting outstanding results for image classification in several different application domains. However, it is also essential to observe that many works reviewed here were developed at the beginning of the pandemic. Many of these works used transfer learning, taking advantage of pre-trained weights produced from other datasets, in general, not composed exclusively of medical images. In addition, many works did not perform fine-tuning. It is easy to understand this kind of strategy in the initial phase of the pandemic, as the data was scarce. However, we conjecture that there is room for further investigations considering the development of studies focused on obtaining more qualified features specifically for COVID-19 detection.

Another important aspect is the imbalance between the number of works developed by using CXR and CT scans. As described in Section 4.4, many more works are devoted to CXR images. Even though CT scan provides a more precise result, it is important to remember that CXR is cheaper and more widespread. In many less economically developed places, CT scan is not even available. So, investigating both scenarios is essential and must continue for different reasons.

Figure 5 displays a word cloud summarizing the most frequent words in the abstracts of all papers. Despite being an informal analysis, the disparity of deep learning when compared to shallow methods is relatively straightforward; the terms referring to deep learning, such as deep, learning, convolutional, CNN, neural, and network, are evident in the word cloud, whereas no visible terms are referring to shallow methods. Another visible difference is the type of image; the terms related to chest X-ray, such as xray and CXR, are more prominent than terms referring to CT scan.

Last but not least, we discuss the feasibility of applying the strategies described in the papers reviewed here in a real scenario. None of the papers reviewed here has been applied in a real scenario, even considering the work that health professionals contributed as co-author. Only three made an application available online, aiming to provide a system that could help support COVID-19 diagnosis. In addition, 40 out of the 100 papers counted on the support of health professionals. In this case, we adopted a quite flexible requirement to define papers with a contribution of health professionals: every paper with at least one co-author affiliated with a hospital, health institute, department, or university, was considered in this category. Thus, despite the impressive progress already made, there are still some important aspects to be addressed in future research.

## Figures and Tables

**Figure 1 sensors-22-07303-f001:**
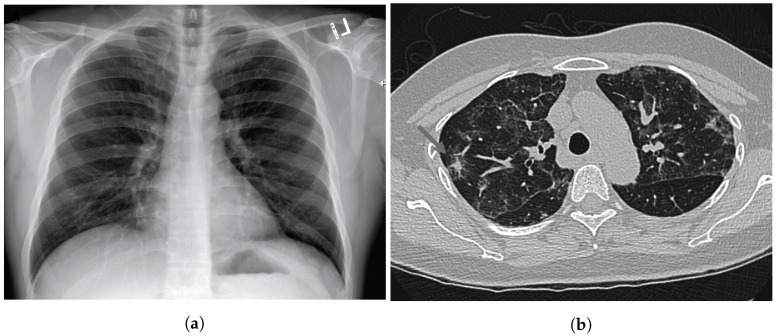
Thoracic medical imaging. (**a**) Example of CXR taken from [1]. (**b**) Example of CT scan taken from [1].

**Figure 2 sensors-22-07303-f002:**
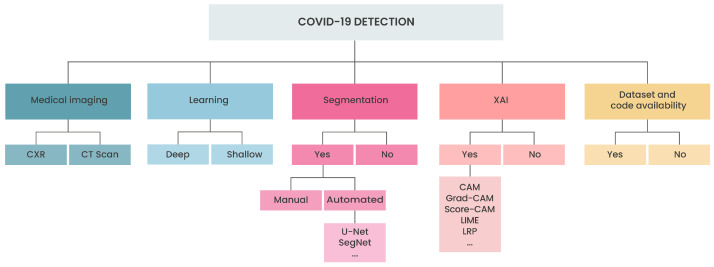
Taxonomy used to conduct the review.

**Figure 3 sensors-22-07303-f003:**
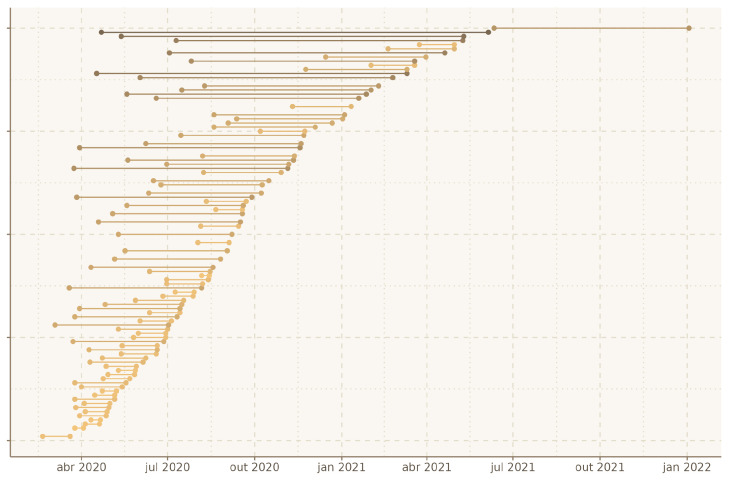
Submission and publication dates.

**Figure 4 sensors-22-07303-f004:**
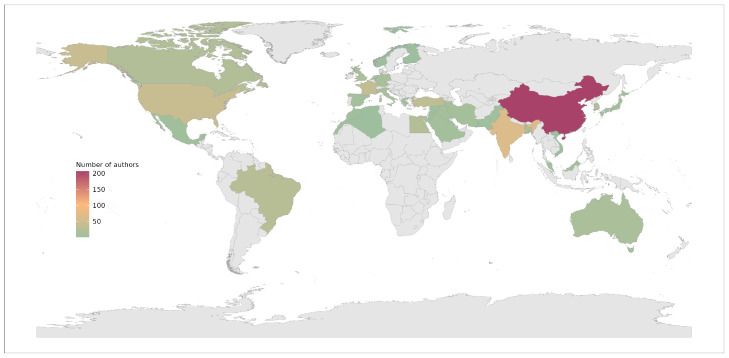
Distribution of authors by country.

**Figure 5 sensors-22-07303-f005:**
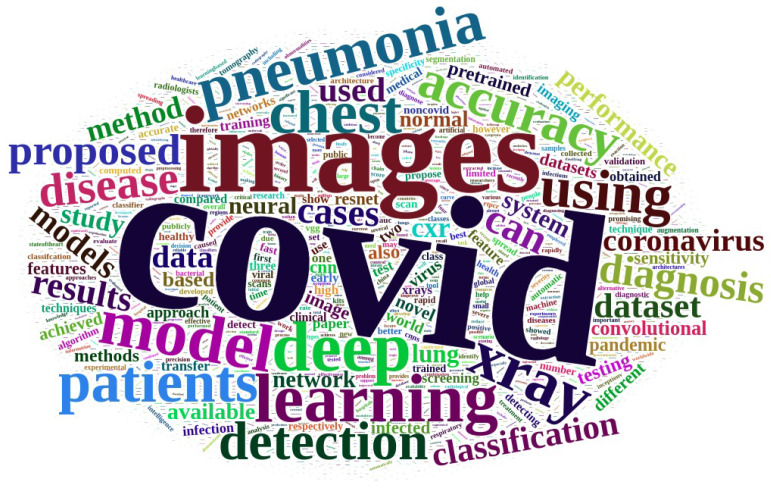
Wordcloud of all abstracts.

**Table 1 sensors-22-07303-t001:** Details of top 100 after each round of filtering.

	Average Number of Citations ^1^	H-Index	Maximum Number of Citations	Minimum Number of Citations
First round	299	95	1848	87
After F1	289	90	1848	80
After F2	251	81	1848	65

^1^ “Average number of citations” corresponds to the total sum of citations obtained by the papers divided by the number of papers.

**Table 2 sensors-22-07303-t002:** Details about the 25 most cited papers.

Rank	Authors-Reference	Year	Citations	CPD ^1^	CT/CXR	Deep/Shallow Learning	Detection/Classification ^2^	Open Code
1	Wang et al. [11]	2020	1848	3.04	CXR	Deep	Both	yes
2	Ozturk et al. [12]	2020	1523	1.89	CXR	Deep	Both	yes
3	Apostolopoulos et al. [13]	2020	1456	1.75	CXR	Deep	Both	no
4	Narin et al. [14]	2021	1275	2.97	CXR	Deep	Both	yes
5	Wang et al. [15]	2021	1121	2.23	CT	Deep	Detection	no
6	Xu et al. [16]	2020	1107	1.49	CT	Deep	Both	no
7	Khan et al. [17]	2020	698	0.91	CXR	Deep	Detection	yes
8	Abbas et al. [18]	2021	640	0.95	CXR	Deep	Classification	yes
9	Song et al. [19]	2021	605	1.24	CT	Deep	both	yes
10	Oh et al. [20]	2020	500	0.63	CXR	Deep	Detection	yes
11	Ardakani et al. [21]	2020	498	0.62	CT	Deep	Detection	no
12	Chen et al. [22]	2020	466	0.76	CT	Deep	Detection	yes
13	Ucar and Korkmaz [23]	2020	465	0.57	CXR	Deep	Both	no
14	Afshar et al. [24]	2020	411	0.62	CXR	Deep	Detection	yes
15	Panwar et al. [25]	2020	349	0.45	CXR	Deep	Both	no
16	Huang et al. [26]	2020	341	0.40	CT	Deep	None	no
17	Togaçar et al. [27]	2020	333	0.42	CXR	Shallow	Classification	yes
18	Pereira et al. [28]	2020	327	0.41	CXR	Shallow	Classification	yes
19	Wang et al. [29]	2020	326	0.46	CT	Both	Both	no
20	Maghdid et al. [30]	2021	311	0.37	Both	Deep	Detection	no
21	Brunese et al. [31]	2020	308	0.41	CXT	Deep	Both	no
22	Loey et al. [32]	2020	298	0.37	CXR	Deep	Both	no
23	Islam et al. [33]	2020	292	0.42	CXR	Deep	Classification	no
24	Ismael and Sengür [34]	2021	291	0.45	CXR	Both	Detection	no
25	Amyar et al. [35]	2020	281	0.44	CT	Deep	Classification	no

^1^ Average number of citations per day starting from the date when the paper was published. ^2^ ‘Detection’ stands for binary classification, and “classification” stands for multi-class.

**Table 3 sensors-22-07303-t003:** Submission and publication dates.

Quarter	Submissions	Publications
2020 Q1	14	1
2020 Q2	46	25
2020 Q3	18	33
2020 Q4	4	18
2021 Q1	3	16
2021 Q2	1	6
2021 Q3	-	-
2021 Q4	-	-
2022 Q1	-	1

**Table 4 sensors-22-07303-t004:** CXR vs. CT scan.

Image Type	Quantity	Average Number of Citations
CT	28	251
CXR	61	269
Both	11	155

**Table 5 sensors-22-07303-t005:** Datasets frequently used to compose image collections.

Dataset	Quantity	Average Number of Citations
cohen ^1^	55	236
kaggle (pneumonia) ^2^	29	206
chestX-ray8/chestX-ray14	22	280
sirm	16	219
radiopaedia	13	301
covid-ct	13	135
rsna	12	282
kaggle covid-19 ^3,4^	8	126
kermany	7	323
covidx	6	520
figure1 ^5^	4	143
sars-cov-2 ct-scan ^6^	4	124

^1^https://github.com/ieee8023/covid-chestxray-dataset (accessed on 12 July 2022); ^2^
https://www.kaggle.com/datasets/paultimothymooney/chest-xray-pneumonia (accessed on 12 July 2022); ^3^
https://www.kaggle.com/datasets/tawsifurrahman/covid19-radiography-database (accessed on 12 July 2022); ^4^
https://www.kaggle.com/datasets/prashant268/chest-xray-covid19-pneumonia (accessed on 12 July 2022); ^5^
https://github.com/agchung/figure1-covid-chestxray-dataset (accessed on 12 July 2022); ^6^
https://www.kaggle.com/datasets/plameneduardo/sarscov2-ctscan-dataset (accessed on 12 July 2022).

**Table 6 sensors-22-07303-t006:** Data privacy.

Data Privacy	Quantity	Average Number of Citations
Public	85	362
Private	15	232

**Table 7 sensors-22-07303-t007:** Classifier type.

Classifier Type	Quantity	Average Number of Citations
Deep	81	279
Shallow	12	139
Both	7	128

**Table 8 sensors-22-07303-t008:** Feature extraction.

Feature Type	Quantity	Average Number of Citations
Deep	8	131
Handcrafted	5	108
Both	6	173

**Table 9 sensors-22-07303-t009:** Transfer learning.

Transfer Learning	Quantity	Average Number of Citations
Yes	65	229
No	32	305
Both	2	182
Not informed	1	130

**Table 10 sensors-22-07303-t010:** Data augmentation.

Data Augmentation	Quantity	Average Number of Citations
Yes	48	243
No	51	263
Both	1	80

**Table 11 sensors-22-07303-t011:** Segmentation strategy.

Segmentation Strategy	Quantity	Average Number of Citations
None	75	239
Manually	2	810
Automated	23	244

**Table 12 sensors-22-07303-t012:** Segmentation strategy.

XAI Method	Quantity	Average Number of Citations
None	75	251
CAM	4	186
Grad-CAM	17	210
Score-CAM	1	211
Saliency maps	1	147
LIME	1	113
Layer-wise Relevance Propagation (LRP)	1	107
GSInquire	1	1848
uAI Intelligent Assistant Analysis System	1	75

**Table 13 sensors-22-07303-t013:** List of the 18 papers not peer reviewed, excluded in F2.

	Authors–Reference	Preprint Repository	Year ^1^	Citations ^2^
1	Hemdan et al. [49]	arXiv	2020	809
2	Gozes et al. [50]	arXiv	2020	726
3	Zheng et al. [51]	MedRxiv	2020	526
4	Shan et al. [52]	arXiv	2020	510
5	Zhang et al. [53]	arXiv	2020	365
6	Farooq et al. [54]	arXiv	2020	349
7	Ghoshal et al. [55]	arXiv	2020	320
8	He et al. [56]	medrxiv	2020	251
9	Hall et al. [57]	arXiv	2020	202
10	Punn et al. [58]	MedRxiv	2020	178
11	Khalifa et al. [59]	arXiv	2020	145
12	Mahdy et al. [60]	MedRxiv	2020	129
13	Alom et al. [61]	arXiv	2020	116
14	Mangal et al. [62]	arXiv	2020	114
15	Kumar et al. [63]	MedRxiv	2020	107
16	Rajinikanth et al. [64]	arXiv	2020	104
17	Gozes et al. [50]	arXiv	2020	93
18	Castiglioni et al. [65]	MedRxiv	2020	76

^1^ Considering the publication date. ^2^ According to Google Scholar on 12 July 2022.

## Data Availability

Not applicable.

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
