# Peer review of "COVID-19 Detection on Chest X-ray and CT Scan: A Review of the Top-100 Most Cited Papers"

_sensors, 2022, doi:10.3390/s22197303_

Round 1
Reviewer 1 Report
This paper studies the top 100 most cited papers in the field of COVID-19 detection. The review is well organized, but I think there are some issues that need further improve before it can be published.
1. Fig. 2 of this paper shows the methods of classification, some of the methods in the figure are in abbreviated form, and the full names and explanations of related concepts, such as CAM and Grad-CAM are not given in the paper. The authors had better give more details about the above methods.
2. Some of the terms proposed in this review are not explained in the previous section, but are explained in Section 4, such as ‘shallow methods’. The authors had better give some explanations when these terms appear in the beginning.
3. The detailed work and main contributions of the 25 most cited papers are described in Section 3, but the relevance of these papers is not highlighted. Does the work in these papers has some relevance with each other? The authors had better introduce more on the features of each cited method.
4. My major concern is that the review may lack of comparison and analysis of experimental results between the literatures. What datasets are these experiments performed on? I think the authors had better performed more comparisons and corresponding analyses for the experimental results.
Author Response
This paper studies the top 100 most cited papers in the field of COVID-19 detection. The review is well organized, but I think there are some issues that need further improve before it can be published.
Thank you for your encouraging words!
We worked hard aiming to attend all your valuable suggestions.
- Fig. 2 of this paper shows the methods of classification, some of the methods in the figure are in abbreviated form, and the full names and explanations of related concepts, such as CAM and Grad-CAM are not given in the paper. The authors had better give more details about the above methods.
First of all, thank you very much for taking the time to review our work.
We adapted the following sentences in Section 2. “Study design and taxonomy” for clarification:
“We evaluated five aspects: i) medical image, chest X-ray (CXR) or computed tomography (CT scan); ii) learning approach, deep or shallow (we use the term shallow method to refer to any method other than deep learning); iii) segmentation strategy, manual or automated using a deep network, common deep strategies includes U-Net [5], SegNet [6], and others; iv) explainable artificial intelligence (XAI), common strategies includes class activation maps (CAM) [7], gradient weighted CAM (Grad-CAM) [8], local agnostic linear model (LIME) [9], layer-wise relevance propagation (LRP) [10], and others; and, v) dataset and code availability.“
- Some of the terms proposed in this review are not explained in the previous section, but are explained in Section 4, such as ‘shallow methods’. The authors had better give some explanations when these terms appear in the beginning.
Thank you for pointing that out!
We believe that by correcting the previous comment we also clarified this one.
- The detailed work and main contributions of the 25 most cited papers are described in Section 3, but the relevance of these papers is not highlighted. Does the work in these papers has some relevance with each other? The authors had better introduce more on the features of each cited method.
Assuming that the number of citations is a metric for scientific quality and importance, it is interesting to analyze the most cited papers in order to find out exactly what they proposed and evaluated to achieve such popularity. The idea of selecting the top 25 most cited papers is purely quantitative and does not take into account any specific highlight. Thus, the Section simply describes each paper, its proposed method and results.
We adapted the following sentences in Section 3. “Overview of Top-25 most cited papers” for clarification:
“This section describes the main highlights of the top-25 most cited papers. We decided to restrict the number of works detailed, aiming to keep it as short as possible while emphasizing its most important contributions. The selection of the top-25 most cited papers is purely quantitative and does not consider any particular characteristic. Assuming that the number of citations is a metric for scientific quality and importance, it is interesting to describe the most cited papers to find out exactly what they proposed and evaluated to achieve popularity in such a short term.“
- My major concern is that the review may lack of comparison and analysis of experimental results between the literatures. What datasets are these experiments performed on? I think the authors had better performed more comparisons and corresponding analyses for the experimental results.
We acknowledge that it would be interesting to compare the experimental results across all selected papers. That was one of our objectives as well. However, as discussed in Section 4.5, almost all papers composed a different dataset in an ad hoc manner for the experimental evaluation which makes it very hard and in some cases unfair to directly compare their recognition rates indiscriminately. The lack of a definitive COVID-19 CXR dataset is a major limitation in almost every work reviewed.
Regarding the datasets used in the experiments, we also discussed them in Section 4.5. Specifically, the Table 5 displays the exact distribution of datasets used across the selected works and their frequency. As some papers used more than one dataset in the evaluation, the summed frequency surpasses 100.
Reviewer 2 Report
-The paper should be interesting ;;;
-it is a good idea to add a block diagram of the proposed research (step by step);;;
-What is the result of the research/paper?;;
-figures should have high quality;;;
-labels of figures should be bigger;;;;
-Fig 5 is not necessary;;;
-what will society have from the paper?;;
-please compare advantages/disadvantages of different approaches;;;
-Conclusion: point out what have you done;;;;
-please add some sentences about future work;;;
Author Response
-The paper should be interesting ;;;
Thank you so much!
-it is a good idea to add a block diagram of the proposed research (step by step);;;
We appreciate your suggestion, however, we are afraid it wouldn’t be appropriate, as it is not a research paper, but a review paper.
-What is the result of the research/paper?;;
We appreciate your suggestion, however, we are afraid it wouldn’t be appropriate. As aforementioned, it is not a research paper, but a review paper.
-figures should have high quality;;;
Thank you for your suggestion.
The only image with a somewhat lower quality is Figure 5, in order to solve this issue we updated it to a more quality version.
Now all figures have a standard resolution of 1000x600 or more and a 300 dpi minimum.
-labels of figures should be bigger;;;;
Thank you for your suggestion. We tried to follow the recommendations described in MDPI instructions for authors, in section ‘Preparing figures, schemes and tables’, which points: ‘All Figures, Schemes and Tables should have a short explanatory title and caption’.
-Fig 5 is not necessary;;;
We do respect your opinion. However, we decided to keep that figure once it is widely used in review papers as a visual way to show the frequency of terms in the field investigated.
-what will society have from the paper?;;
Since the beginning of the COVID-19 pandemic, many works have been published proposing solutions to the problems that arose in this scenario. In this vein, one of the topics that attracted more attention is the development of computer-based strategies to detect COVID-19 from thoracic medical imaging, such as chest x-ray (CXR) and computerized tomography scan (CT scan). By searching for works already published on this theme, we can easily find thousands of them. However, it is not a trivial task to identify the most promising works considering their impact on the research community. By developing this review paper, we tried to contribute to the research community, organizing a discussion on this topic, and bringing the most prominent information about in a summarized form. By contributing to the research community, we believe that we are contributing to scientific progress and to technological development, which indirectly will benefit society.
-please compare advantages/disadvantages of different approaches;;;
Since the paper is categorized as a review paper, we tried to explore the literature and to get the most relevant details of the selected works taking into account the aspects described in Section 2 (‘Study design and taxonomy’). In our point of view, this strategy is more appropriate in this type of paper.
-Conclusion: point out what have you done;;;;
Thank you for your valuable suggestion!
Aiming to make clearer what we have done also in the ‘Concluding remarks’ section, we introduced the following paragraph in that section:
“Based on the rationale that the number of citations obtained by a paper is probably the most straightforward and intuitive way to verify its impact on the research community, we described here a review on the top-100 most cited papers considering the development of computer-based strategies for COVID-19 detection from thoracic medical imaging. Following, we highlight some remarkable findings and we analyze them from the different perspectives addressed in this review.”
-please add some sentences about future work;;;
In our opinion, the main aspects to be covered in future works, refer to the evaluation of the proposed methods in more realistic scenarios. In addition, it would be desirable if future works could have a more representative participation of health professionals, as we already described in the ‘Concluding remarks’ section.
Reviewer 3 Report
The article screened papers on the use of lung medical imaging to diagnose COVID-19 in the past two years, and briefly introduced the most cited papers. A method for screening papers from various aspects is designed, and the number of citations is used as the main evaluation criterion, and 25 papers that are most helpful to the industry are screened for introduction. Compared with other reviews, the author shows his work on paper screening, and analyzes some statistical data of these papers, so that researchers who read this review can quickly find their interesting direction and learn from the corresponding articles. It’s helpful to the community.
Major:
(1) This paper mainly assess the machine learning –based methods. Maybe the title of this paper gives a more accurate subset is better , (with machine learning ),such s “a review of the top-100 most cited papers using machine learning” …
(2) About the statistic data, the unit is by paper number ,not consider the paper cited counts. Or consider the weight of papers based on cited counts
Minor:
(1) Table 1. The average number of citaions, what’s the denominator after the filtering?
Same ?
(2) Line 449, authors? or papers?
Author Response
The article screened papers on the use of lung medical imaging to diagnose COVID-19 in the past two years, and briefly introduced the most cited papers. A method for screening papers from various aspects is designed, and the number of citations is used as the main evaluation criterion, and 25 papers that are most helpful to the industry are screened for introduction. Compared with other reviews, the author shows his work on paper screening, and analyzes some statistical data of these papers, so that researchers who read this review can quickly find their interesting direction and learn from the corresponding articles. It’s helpful to the community.
Thank you for your encouraging words!
We worked hard aiming to attend all your valuable suggestions.
Major:
(1) This paper mainly assess the machine learning –based methods. Maybe the title of this paper gives a more accurate subset is better , (with machine learning ),such s “a review of the top-100 most cited papers using machine learning” …
Thank you so much for your suggestion!
We agree that the inclusion of the term ‘machine learning’ in the title would make it more precise. However, it is not so trivial to include it keeping, at the same time, the title appropriately concise. Thus, we are presenting to the editor the following alternatives for the title, and we intend to discuss with him about the convenience of each option:
- COVID-19 detection on chest X-ray and CT scan: a review of the top-100 most cited papers (this is the original title)
- COVID-19 detection on chest X-ray and CT scan using machine learning: a review of the top-100 most cited papers
- COVID-19 detection on chest X-ray and CT scan: a machine learning focused review of the top-100 most cited papers
(2) About the statistic data, the unit is by paper number ,not consider the paper cited counts. Or consider the weight of papers based on cited counts
We are not sure if we have properly understood your comment, we are sorry for that!
Anyway, if you refer to Table 1, following we describe each column in details:
- ‘Average number of citations’: corresponds to the total sum of citations obtained by the top-100 papers divided by the number of papers. The average decreases after each filter round, as some papers are excluded and substituted by other less cited papers.
- ‘H-index’: corresponds to the h-index obtained from the top-100 paper. Again, the h-index decreases after each filter round, as some papers are excluded and substituted by other less cited papers.
- ‘Maximum number of citations’: corresponds to the number of citations obtained by the most cited paper (the top-one). This number doesn’t change, as the top cited paper was not excluded in any filter round.
- ‘Minimum number of citations’: corresponds to the number of citations of the hundredth paper. This number decreases after each filter round as less cited papers are taken to compose the top-100 to replace excluded papers.
Minor:
(1) Table 1. The average number of citaions, what’s the denominator after the filtering?
Same ?
Thank you for pointing that out!
Table 1 displays the average number of citations before and after the application of each filter. The denominator remains the same, i.e., the number of papers. The average number of citations reduced after filtering due to some highly cited papers being filtered out.
In order to make it clearer, we introduced a footnote on Table 1, explaining that the ‘average number of citations’ corresponds to the total sum of citations obtained by the papers divided by the number of papers.
(2) Line 449, authors? or papers?
The number and percentages refers to authors, since a paper can have multiple authors with different nationalities, we decided to analyze each author individually.
Round 2
Reviewer 1 Report
All reviewer's comments in the first round have been addressed in the current version. I recommend the paper to be accepted for publication.
Author Response
Thank you so much for taking the time to review our paper!
Reviewer 2 Report
It is a good idea to add some photos of measurements, detection, and application of the research/paper.
Relation to Sensors should be added; - photo, figure
Author Response
1. It is a good idea to add some photos of measurements, detection, and application of the research/paper.
The paper is a literature review and not an original proposal; hence we do not have measurements, detection, or application outside of the general statistics presented and discussed in Section 3.
2. Relation to Sensors should be added; - photo, figure
Thank you for your suggestion.
Unfortunately, we believe the paper already contains too many figures and tables, and adding more would probably make the article too lengthy. Moreover, we also think that the relation to Sensors is clear; we are submitting the current review to the "COVID-19 Biosensing Technologies" Special Issue, which clearly states that reviews are also part of the scope: "Research articles, review articles as well as short communications are invited"
Reviewer 3 Report
The quanity of papers, for exmaple, in Table 4. CT or CXR was used as the experiment material in the top100 papers. But Citation Per Paper is different, and the importance of each paper is different. It's not just adding quantities of papers. It's better add new column in these tables.
===============
(2) About the statistic data, the unit is by paper number ,not consider the paper cited counts. Or consider the weight of papers based on cited counts
We are not sure if we have properly understood your comment, we are sorry for that!
Anyway, if you refer to Table 1, following we describe each column in details:
- ‘Average number of citations’: corresponds to the total sum of citations obtained by the top-100 papers divided by the number of papers. The average decreases after each filter round, as some papers are excluded and substituted by other less cited papers.
- ‘H-index’: corresponds to the h-index obtained from the top-100 paper. Again, the h-index decreases after each filter round, as some papers are excluded and substituted by other less cited papers.
- ‘Maximum number of citations’: corresponds to the number of citations obtained by the most cited paper (the top-one). This number doesn’t change, as the top cited paper was not excluded in any filter round.
- ‘Minimum number of citations’: corresponds to the number of citations of the hundredth paper. This number decreases after each filter round as less cited papers are taken to compose the top-100 to replace excluded papers.
Author Response
Thank you for your suggestion, now we understood your comment, we added a column displaying the average number of citations in Tables 4, 5, 6, 7, 8, 9, 10, 11 and 12.